# Optimal Time and Target for Evaluating Energy Delivery after Adjuvant Feeding with Small Bowel Enteral Nutrition in Critically Ill Patients at High Nutrition Risk

**DOI:** 10.3390/nu11030645

**Published:** 2019-03-16

**Authors:** Wei-Ning Wang, Mei-Fang Yang, Chen-Yu Wang, Chiann-Yi Hsu, Bor-Jen Lee, Pin-Kuei Fu

**Affiliations:** 1Department of Food and Nutrition, Taichung Veterans General Hospital, Taichung 40705, Taiwan; sherry@vghtc.gov.tw (W.-N.W.); mfyang@vghtc.gov.tw (M.-F.Y.); 2Department of Critical Care Medicine, Taichung Veterans General Hospital, Taichung 40705, Taiwan; chestmen@gmail.com (C.-Y.W.); borjenlee@gmail.com (B.-J.L.); 3Department of Nursing, Hungkuang University, Taichung 43302, Taiwan; 4Biostatistics Task Force of Taichung Veterans General Hospital, Taichung 40705, Taiwan; chiann@vghtc.gov.tw; 5School of Medicine, Chung Shan Medical University; Taichung 40201, Taiwan; 6School of Chinese Medicine, China Medical University, Taichung 43302, Taiwan; 7College of Human Science and Social Innovation, Hungkuang University, Taichung 43302, Taiwan; 8Science College, Tunghai University, Taichung 40704, Taiwan

**Keywords:** critically ill patients, energy delivery, high nutrition risk, modified nutric score, small bowel enteral nutrition

## Abstract

Small bowel enteral nutrition (SBEN) may improve nutrient delivery to critically ill patients intolerant of gastric enteral nutrition. However, the optimal time and target for evaluating SBEN efficacy are unknown. This retrospective cohort study investigates these parameters in 55 critically ill patients at high nutrition risk (modified NUTRIC score ≥ 5). Daily actual energy intake was recorded from 3 days before SBEN initiation until 7 days thereafter. The energy achievement rate (%) was calculated as follows: (actual energy intake/estimated energy requirement) × 100. The optimal time was determined from the day on which energy achievement rate reached >60% post-SBEN. Assessment results were as follows: median APACHE II score, 27; SOFA score, 10.0; modified NUTRIC score, 7; and median time point of SBEN initiation, ICU day 8. The feeding volume, energy and protein intake, and achievement rate (%) of energy and protein intake increased significantly after SBEN (*p* < 0.001). An energy achievement rate less than 65% 3 days after SBEN was significantly associated with increased mortality after adjusting for confounding factors (odds ratio, 4.97; 95% confidence interval, 1.44–17.07). SBEN improves energy delivery in critically ill patients who are still at high nutrition risk after 1 week of stomach enteral nutrition.

## 1. Introduction

Gastrointestinal motility disorder is a common problem among critically ill patients, independent of morbidity and mortality [1,2]. Up to 60% of critically ill patients in intensive care units (ICU) are reported to experience gastrointestinal dysmotility, including delayed gastric emptying, abnormal motility patterns, and impaired intestinal barrier integrity requiring therapeutic intervention [2,3]. Current guidelines suggest the use of a screening tool such as the nutrition risk in the critically ill (NUTRIC) score or modified NUTRIC score to determine nutrition risk in ICU patients to initiate early enteral nutrition (EN) therapy [4,5,6,7]. Patients with an mNUTRIC score ≥5 are defined as ‘high nutrition risk’ and require additional energy intake to reduce mortality [4,5,8]. Our previous study showed that an energy achievement rate ([actual energy intake/estimated energy requirement] × 100) below 65% in critically ill patients was associated with an increased risk of mortality (odds ratio [OR], 1.6%; 95% confidence interval [CI], 1.01–2.47) [1]. Our previous study also indicates that targeted energy intake is an important predictor of mortality predictor in critically ill patients with high nutrition risk [9]. Previous systematic reviews and meta-analysis have shown that the benefits of early EN include decreased in-hospital mortality, infectious-disease morbidity, and incidence of pneumonia [10,11,12,13].

The recommended route for early EN is through the stomach, as this route is technically easier and thus may facilitate earlier initiation of EN [4,5]. However, small bowel EN (SBEN) has theoretical advantages over stomach EN because it has lower risks of gastroesophageal reflux and aspiration pneumonia. A previous multicenter randomized controlled trial (RCT) comparing gastric to small bowel EN in critically ill patients reported no difference in clinical outcomes between routes, including mortality, hospital days, and incidence of pneumonia [14]. Other studies observed similar results, reporting that SBEN provides greater nutrient delivery [14,15,16] and has a lower risk of pneumonia [16,17,18] than does gastric EN.

The American Society for Parenteral Enteral Nutrition (ASPEN) and the Society of Critical Care Medicine (SCCM) and European Society for Parenteral Enteral Nutrition (ESPEN) guidelines recommend that the level of infusion be diverted lower in the gastrointestinal tract in critically ill patients at high risk for aspiration or intolerance to gastric EN [4,5,6]. For patients who are unable to meet >60% of the energy and protein requirements by the enteral route alone, the use of supplemental parenteral nutrition (PN) should be considered after 7–10 days of EN [4,5]. However, when to shift the route of gastric EN to SBEN before PN is unclear, as is the optimal time for evaluating SBEN sufficiency in these patients. This retrospective cohort study aims to determine the optimal time for initiating SBEN, the critical time by which >60% of the energy goal should be achieved after SBEN and the effect of nutrition target achievement on mortality in critically ill patients at high nutrition risk.

## 2. Materials and Methods

### 2.1. Study Design

This retrospective, observational cohort study was conducted in the respiratory intensive care unit of Taichung Veterans General Hospital, a tertiary medical center located in central Taiwan between January 2014 and December 2015. The study protocol was fully reviewed and approved by the Institutional Review Board of Taichung Veterans General Hospital (IRB number, CE16028A; date of approval, 29 January 2016), which also waived the requirement for informed patient consent due to the retrospective study design and the availability of all de-linked data via an electronic health record system.

### 2.2. Patient, ICU Setting and Nutrition Routine

The respiratory ICU has 24 beds and admits all adult medical patients, 70% of who were diagnosed with lower respiratory tract infection-related critical illness, including sepsis, septic shock, acute respiratory failure, and acute respiratory distress syndrome. Nutrition evaluation in the respiratory ICU is covered by a registered dietitian. The volume-based feeding protocol has been implemented in TVGH’s ICU since June 2015 as our previous report [1]. The modified NUTRIC score, nutrition goal, and energy target achievement rate are routinely determined by the dietitian for every respiratory ICU patient. The mNUTRIC score used in this study has 5 components, including (1) age, (2) acute physiology and chronic health evaluation II (APACHE II) score, (3) baseline simplified organ failure assessment (SOFA) score, (4) number of comorbidities, and (5) days in hospital before ICU admission [7]. The mNUTRIC score (0–9 points) [7,19] was calculated after ICU admission and recorded in the electronic database. The energy achievement rate (%) was calculated as follows: (actual energy intake/estimated energy requirement) × 100 [1]. The energy requirement was determined using the recommendation of 25–30 kcal/kg/day, and protein intake was monitored as 1.2–2 g protein/kg body weight/day according to the 2016 ASPEN/SCCM guidelines [4,5]. Early EN was initiated in all critically ill patients on the first day in the respiratory ICU except in special circumstances. On the 7th ICU day, a gastroenterologist was consulted to place a transpyloric tube for small bowel enteral nutrition (SBEN) in patients with severe malnutrition (energy target achievement rate <30%). From January 2014 to December 2015, 60 critically ill patients were treated with SBEN in the respiratory ICU. After excluding patients diagnosed with acute pancreatitis (*n* = 1) or who died within 3 days of SBEN commencement (*n* = 4), a total of 55 patients at high nutrition risk remained in the cohort for analysis.

### 2.3. Data Collection, Assessment, and Outcome Measures

Data were collected regarding age; sex; body mass index (BMI); and clinical outcomes, including severity of illness (APACHE II score at the 1st ICU day and SOFA score at the 1st, 3rd and 7th ICU days), major comorbidities, length of ventilator dependence, length of hospital and ICU stays, laboratory data on the first ICU day, and mortality in the ICU or hospital. The index day was the day of transpyloric tube insertion for SBEN. Daily energy intake was recorded for stomach EN for the 3 days before the index day and from SBEN on the index day and 7 days thereafter. The amount of actual nutrition delivery is based on nursing documentation. We defined the average value of the 3 days before index day as ‘before SBEN.’ The daily energy intake was compared before and after SBEN. The feeding efficiency was evaluated using the following 5 parameters: (1) actual feeding volume (mL/day); (2) actual energy intake (kcal/day); (3) energy intake achievement rate (%); (4) actual protein intake (g/day); and (5) protein intake achievement rate (%). The primary endpoint was a comparison of daily feeding efficiency before and after SBEN. The optimal time was determined as the number of days after SBEN initiation that the energy achievement rate surpassed 60%. Cox regression analysis was used to determine whether increased energy achievement rates after SBEN are associated with lower in-hospital mortality in this population.

### 2.4. Statistical Analysis

Statistical analysis was conducted using the SPSS statistical software package (version 22.0; International Business Machines Corp, Armonk, NY, USA). Categorical variables were presented as frequency and percent and analyzed using the chi-squared test to determine significance. For nonparametric data distribution, differences between groups were assessed using the Mann–Whitney U test or Wilcoxon signed ranks test, and results are presented as the median and interquartile range (IQR). Cox regression analysis was used to assess factors associated with mortality. The strength of association is presented as the Hazard ratio (HR) and 95% Confidence interval (CI). Receiver operating characteristic (ROC) curves were used to evaluate the discriminative ability of energy intake achievement rates on day 3 after the initiation of small bowel feeding to identify the survival group. All tests were two-sided, with *p* < 0.05 considered significant.

## 3. Results

Table 1 shows the demographic characteristics, severity scores, comorbidities, energy intake, and clinical outcomes of critically ill patients. A total of 55 patients (21 women, 34 men) were retrospectively analyzed in this study. The median mNUTRIC score of this cohort was 7 (IQR, 5–8), and the median albumin level was 2.5 g/dL (IQR, 2.2–2.9), indicating that these patients were in the high nutrition risk. The APACHE II and SOFA scores also indicated the high clinical severity of this cohort. The overall ICU mortality rates and hospital mortality rates were 32.7% and 45.5%, respectively. The most common comorbidities were cardiovascular disease, cancer, chronic obstructive lung disease, diabetes mellitus, uremia, and liver cirrhosis. The median time to SBEN initiation after ICU admission was 8 days. The daily average energy intake achievement rate (%) on days 2–7 after SBEN initiation was 57.6%, 55.9%, 61.6%, 65%, 63.8% and 66.8%, respectively. The actual daily energy parameters of feeding efficiency, including actual feeding volume (mL/day), actual energy intake (kcal/day), actual protein intake (g/day), energy intake achievement rate (%), and protein achievement rate (%) before and after SBEN initiation increased significantly day by day (Figure 1). A significant increase was observed in all feeding parameters on post-SBEN day 3 as compared to those before SBEN (Table 2). The feeding volume increased significantly (2-fold increase; *p* < 0.001). The energy intake, energy achievement rate, and actual protein intake (g/day) all increased 2–3-fold after SBEN initiation (*p* < 0.001) (Table 2).

Surviving and non-surviving patients were compared with respect to all parameters (Table 3). Non-surviving patients had higher mNUTRIC and SOFA scores, longer ICU stays, and lower energy intake achievement rates (%) on each day subsequent to SBEN initiation. In the survival group, the median energy intake reached 60% on days 2–3 post-SBEN. In contrast, the median energy intake achievement in the non-survival group remained below 50%, even on day 4 post-SBEN (Table 3). A receiver operating characteristic (ROC) curve was used to evaluate the discriminative ability of energy intake achievement rates 3 days after SBEN initiation to identify the survival group; the cutoff value was 65% (Appendix A). In the survival group, two-thirds of the patients achieved the energy target of 65%. In contrast, only 20% of the patients in the non-survival group achieved this goal (Table 3).

Results of Cox regression analysis of factors associated with mortality are shown in Table 4. Univariate analysis identified only two factors associated with mortality: SOFA score on the 7th ICU day (HR, 1.11; 95% CI, 1.00–1.24; *p* = 0.047) and energy achievement rates < 65% on post-SBEN day 3 (HR, 4.58; 95% CI, 1.55–13.53; *p* = 0.006). Multivariate analysis was used to investigate factors found to be significant with the univariate analysis in addition to the mNUTRIC score, which was included in the multivariate analysis because of its importance in ICU mortality. After adjustment for confounding factors, multivariate analysis showed that post-SBEN day 3 energy achievement rate <65% (HR, 4.97; 95% CI, 1.44–17.07; *p* = 0.011) was the only predictive factor that differed significantly between the survival and non-survival groups. SBEN day 3 energy achievement rate >65% was associated with lower 30-, 60-, and 90-day mortality after adjustment for age, sex, BMI, and APACHE II and SOFA scores in patients with high nutrition risk (Figure 2).

## 4. Discussion

The present study reports three major findings regarding malnourished, critically ill patients. First, after 7 days of stomach EN, the initiation of adjuvant SBEN on day 8 improved energy delivery. Second, the feeding efficiency after SBEN initiation increased significantly and reached the nutritional goal in 3 days. Third, achieving 65% of the energy requirements by day 3 after SBEN initiation was the key factor associated with survival.

All patients in this cohort had high mNUTRIC scores (median mNUTRIC score, 7; IQR, 5–8) and experienced low energy achievement rates even though gastric feeding was started early, within 48 h of ICU admission. The current ASPEN/SCCM and ESPEN guidelines recommend that nutrition support therapy from early EN should be initiated within 24–48 h in the critically ill patient who is unable to maintain volitional intake [4,5,6]. Accordingly, early EN from gastric access should be used as the standard approach and post-pyloric feeding should be used in patients with gastric feeding intolerance not solved with prokinetic agents [4,5,6]. Since current evidence does not indicate routine feeding via SBEN [14], the right time to shift from gastric EN to SBEN has been unclear. Our observation that SBEN initiation on day 8 improves energy delivery could be a choice of feeding route for critically ill ICU patients with high nutrition risk. We also demonstrated SBEN could improve the feeding efficacy for malnutrition patients with a high mNUTRIC score, therefore, further study is warranted to determine how early is possible to initiate SBEN if failure to achieve 60% of feeding goal in ICU. We suggest that for medical ICU patients with high mNUTRIC scores, adjuvant SBEN as early as possible is another option before shifting to parenteral nutrition.

We also observed that SBEN significantly increased the feeding efficiency, reaching the nutritional goal in 3 days. Previous studies demonstrated greater feeding efficiency with small bowel feeding as compared to gastric feeding [14,15,16,18]. However, for critically ill patients with high nutrition risk who fail to achieve the nutritional target after gastric tube feeding, the efficacy of SBEN before shifting to parental nutrition (PN) has remained unclear. The current ASPEN/SCCM guidelines recommend that supplemental PN only be considered if >60% of energy and protein requirements are not met after 7–10 days of EN because initiating PN before this time does not improve outcomes [4,5]. Our data suggest that for critically ill patients with mNUTRIC scores ≥5 and poor gastric tube feeding, SBEN for 3 days is another option before PN.

We observed that achieving 65% of the energy requirements within 3 days of SBEN initiation is the key factor associated with survival (OR, 4.97; 95% CI, 1.44–17.07) in malnourished, critically ill patients with high nutrition risk. Previous studies have shown that adequate nutritional support has a significant mortality benefit in critically ill patients with a high risk of malnutrition [20,21]. Suboptimal energy delivery is an important factor associated with increased nosocomial infection, longer duration of ventilator dependency and hospital and ICU stay, and greater mortality [1,20,22]. However, ample evidence indicates that patients with low mNUTRIC scores gain no survival benefit from adequate nutritional support [7,9,21,23,24]. Our previous studies show that high–nutrition-risk ICU patients who receive at least 65% of their energy requirements and consume at least 800 kcal/day have significantly lower in-hospital, 14-day, and 28-day mortality rates [1,9]. In this study, the cut-off value for nutritional intake distinguishing survival from non-survival was 65%, the same as that indicated by our previous study.

This study has several limitations. First, the study was conducted at a single ICU; therefore, the generalizability of the results may be challenged. Second, the retrospective design of the study precludes intervention; thus, the duration of post-pyloric feeding might differ greatly between patients. In addition, we don’t know how early is the optimal time to start SBEN by the retrospective observational cohort study. Therefore, a prospective study design is warranted. In actual practice, the feeding protocols vary between ICUs. This study analyzed data from a single respiratory ICU that has a feeding protocol approved by a registered dietitian. A gastroenterologist was consulted to evaluate transpyloric tube insertion for SBEN for patients with an energy target achievement rate <30% on the 7th ICU day. Therefore, heterogeneity in the care and feeding protocol was minimized in our ICU. Third, many treatments commonly administered in the ICU, such as Computed tomography scans, invasive procedures, and gastrointestinal endoscopy, may interrupt feeding protocols. Finally, the present results may not be generalizable to critically ill patients in a surgical ICU, as this study was conducted only in medical ICU patients.

## 5. Conclusions

SBEN improves energy delivery to those critically ill patients who remain at high nutrition risk after one week of stomach EN. The key factor determining survival in this population is the intake of 65% of the energy requirement within 3 days of SBEN initiation. For medical ICU patients with high mNUTRIC scores, adjuvant SBEN as early as possible is another option before shifting to parenteral nutrition if gastric EN fails to achieve 60% of the energy goal.

## Figures and Tables

**Figure 1 nutrients-11-00645-f001:**
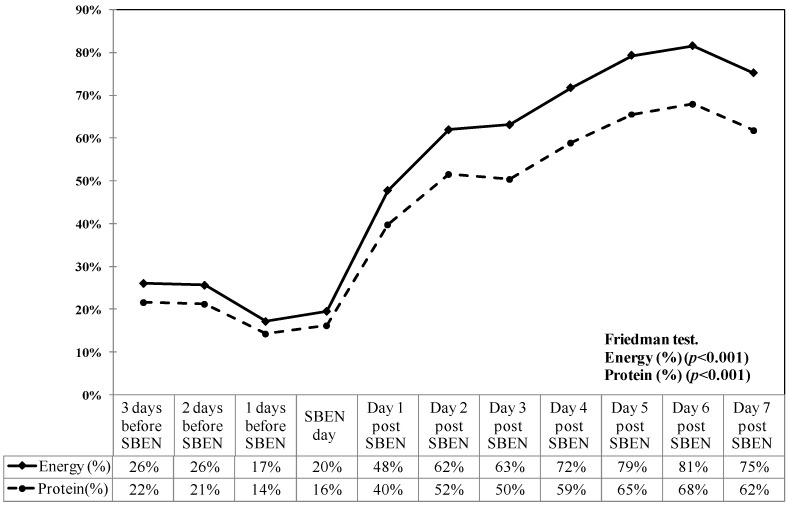
Achievement rate of energy and protein intake before and after small bowel enteral nutrition (SBEN) (*n* = 49).

**Figure 2 nutrients-11-00645-f002:**
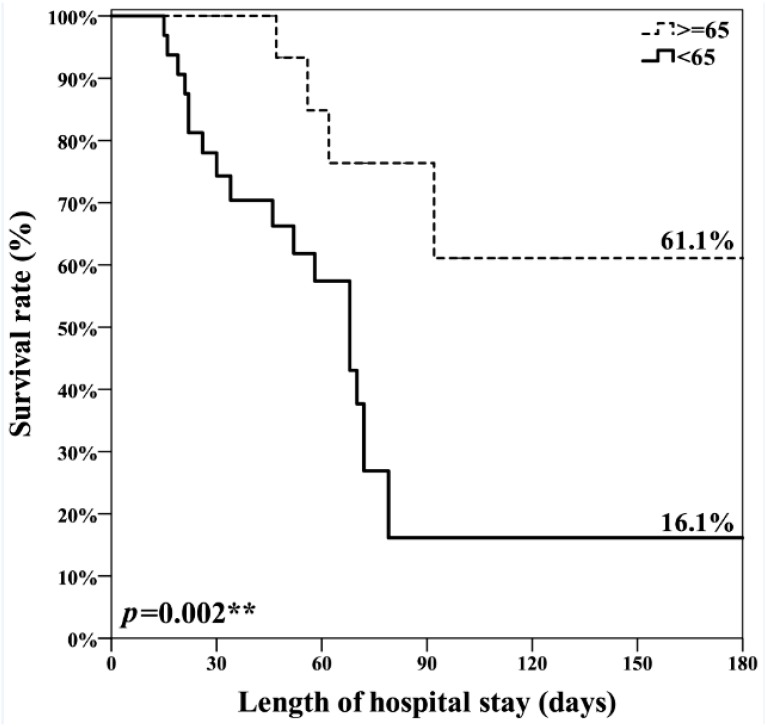
Energy achievement rate > 65% was significantly associated with lower 30-, 60-, and 90-day mortality in patients with high nutrition risk. ** *p* < 0.01.

**Table 1 nutrients-11-00645-t001:** Demographic characteristics, severity scores, comorbidities and clinical outcomes of all malnourished patients receiving small bowel enteral nutrition in the intensive care unit (*n* = 55).

Variables	Median	(IQR)
Age (year)	69.9	(60–79.1)
Sex (Female/Male)	21/34	
Body mass index (kg/m^2^)	23.9	(21.5–27.6)
Albumin (g/dL)	2.5	(2.2–2.9)
mNUTRIC score	7	(5–8)
APACHE II score	27	(24–33)
SOFA day 1	10	(8–13)
SOFA day 3 (*n* = 54)	9	(6–12)
SOFA day 7 (*n* = 53)	8	(5–11.5)
Length of ICU stay (days)	21	(15–32)
Length of Hospital stay (days)	49	(28–72)
Length of ventilatory dependency (days)	27	(17–58)
Day from ICU admission to small bowel feeding	8	(5–13)
Comorbidities (*n*, %)		
Cardiovascular disease	37	(67.3%)
Cancer	24	(43.6%)
Chronic obstructive lung disease	18	(32.7%)
Diabetes mellitus	16	(29.1%)
Uremia	11	(20.0%)
Liver cirrhosis	11	(20.0%)
Average Energy intake achievement rate (%) post SBEN		
Day 2	57.6	(35.8–70.9)
Day 3	55.9	(41.4–76.7)
Day 4	61.6	(41.2–75.3)
Day 5	65.0	(42.9–78.6)
Day 6	63.8	(45.3–81)
Day 7	66.8	(48.9–83.4)
ICU mortality (*n*, %)	18	(32.7%)
In-hospital mortality (*n*, %)	25	(45.5%)

mNUTRIC, modified nutrition risk in the critically ill; APACHE II, Acute Physiology and Chronic Health Evaluation II; SOFA, simplified organ failure assessment; ICU, intensive care unit; SBEN, small bowel enteral nutrition.

**Table 2 nutrients-11-00645-t002:** Nutritional achievement of malnourished critically ill patients before and 3 days after SBEN in malnourished critically ill patients.

Variables	Before SBEN	Average Post-SBEN Day 3
Actual feeding volume (mL/day)	454.0 (244.3–637.3)	948.0 (660.7–1188) **
Actual energy intakes (kcal/day)	409.2 (205.8–559.2)	907.2 (594.6–1187.6) **
Actual protein intakes (g/day)	16.4 (8.2–22.4)	36.3 (23.8–47.7) **
Energy intake achievement rate (%)	25.5 (16–33.9)	55.9 (41.4–76.7) **
Protein intake achievement rate (%)	21.3 (13.3–28.2)	46.6 (33.9–63.9) **

Values are Median (IQR). Wilcoxon Signed Ranks test; ** *p* < 0.01. SBEN, small bowel enteral nutrition.

**Table 3 nutrients-11-00645-t003:** Patient demographic characteristics, severity index, comorbidities, and energy intake achievement rates in survivors and non-survivors.

Variables	Non-Survival (*n* = 25)	Survival (*n* = 30)
Age (year)	72.9 (59.1–78.1)	67.3 (60.2–79.6)
Male (*n*, %)	14 (56.0%)	20 (66.7%)
Body mass index (kg/m^2^)	25.6 (21.1–29.8)	23.6 (21.5–26.1)
Albumin (g/dL)	2.4 (2.1–2.6)	2.5 (2.3–3.1)
mNUTRIC score	7.0 (7–8)	6.5 (5–8) *
APACHE II score	31.0 (24.5–36)	26.0 (23.8–31)
SOFA day 1	11.0 (8.5–13.5)	10.0 (6–12.3)
SOFA-day 3 (*n* = 54)	11.0 (6–12.5)	8.0 (6–11.5)
SOFA-day 7 (*n* = 53)	10.0 (5.5–15)	7.5 (4–10) **
Length of ICU stay (days)	26.0 (19–38)	18.0 (13.8–25.5) *
Length of Hospital stay (days)	56.0 (24–71)	48.0 (28.8–76)
Length of ventilatory dependency (days)	27.0 (19.5–56.5)	23.0 (13.8–65.3)
Day from ICU admission to small bowel feeding	9.0 (5–15.5)	7.0 (4.8–13)
Comorbidities (*n*, %)		
Cardiovascular disease	15 (60.0%)	22 (73.3%)
Cancer	8 (32.0%)	16 (53.3%)
Chronic obstructive lung disease	10 (40.0%)	8 (26.7%)
Diabetes mellitus	5 (20.0%)	11 (36.7%)
Uremia	7 (28.0%)	4 (13.3%)
Liver cirrhosis	7 (28.0%)	4 (13.3%)
Average of Energy intake achievement rate (%) post SBEN		
2nd day	45.9 (26–65.4)	65.2 (43.2–74) *
3rd day	48.2 (27–64.1)	67.1 (49.3–82.5) **
4th days	43.5 (31.5–67)	73.0 (52–86.2) **
5th days	52.4 (29.7–68.9)	73.6 (58.7–88.8) **
6th days	53.1 (30.9–66.6)	72.5 (63.1–91.3) **
7th days	50.3 (31.7–65.2)	75.4 (61.7–92.5) **
Energy intake achievement rate (%) ≥65 (Post SBEN 3rd day) (*n*, %)	5 (20.0%)	18 (60.0%) **

Values are Median (IQR). Mann–Whitney U test. Chi-Square test. * *p* < 0.05, ** *p* < 0.01. mNUTRIC, modified nutrition risk in the critically ill. APACHE II, Acute Physiology and Chronic Health Evaluation II; SOFA, simplified organ failure assessment ICU, intensive care unit. SBEN, small bowel enteral nutrition.

**Table 4 nutrients-11-00645-t004:** Cox regression.

Title 1	Univariate Analysis	Multivariate Analysis
Age (year)	1.01 (0.98–1.05)	
Gender (Female vs. Male)	1.34 (0.60–3.01)	
Body mass index (kg/m^2^)	0.97 (0.88–1.06)	
Albumin (mg/dL)	0.51 (0.25–1.05)	
mNUTRIC score	1.16 (0.85–1.59)	0.95 (0.70–1.30)
APACHE II score	1.02 (0.96–1.07)	
SOFA day 1	1.00 (0.91–1.09)	
SOFA day 3	1.04 (0.94–1.15)	
SOFA day 7	1.11 (1.00–1.24) *	1.02 (0.91–1.15)
Energy intake achievement rate (%) (Post-SBEN day 3) (<65% vs. ≥65%)	4.58 (1.55–13.53) **	4.97 (1.44–17.07) *

Cox regression. * *p* < 0.05, ** *p* < 0.01. mNUTRIC, modified nutrition risk in the critically ill. APACHE II, Acute Physiology and Chronic Health Evaluation II; SBEN, small bowel enteral nutrition.

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
