# Peer review of "Optimal Time and Target for Evaluating Energy Delivery after Adjuvant Feeding with Small Bowel Enteral Nutrition in Critically Ill Patients at High Nutrition Risk"

_nutrients, 2019, doi:10.3390/nu11030645_

Reviewer 1 Report

It would be interesting to determine which type of patients will present intolerance to gastric enteral nutrition so as not to wait 7 days to place the intestinal nutrition.

The discussion is somewhat short and should have more impact on the possibility of early bowel nutrition in patients with high NUTRICscore

Author Response

1. It would be interesting to determine which type of patients will present intolerance to gastric enteral nutrition so as not to wait 7 days to place the intestinal nutrition.

Response:

Initial EN from gastric tube with 24-48 hours is ICU routine by current guideline. Previous study revealed that early SBEN in the first ICU day did not have survival benefit from those critical ill patients with malnutrition. The current ASPEN/SCCM guidelines also recommend that supplemental PN only be considered if >60% of energy and protein requirements are not met after 7–10 days of EN. Our ICU routine will start to evaluate the indication for SBEN at the 5-7th ICU days. Why we start to evaluate SBEN insertion at the 5-7th ICU day. That is because the majority patients in our ICU suffered from septic shock and ARDS needed to resuscitate with large fluid volume. General edema and poor feeding status will at peak status at 3-5th ICU day. GI man will suggest we try to use diuretics and increase the dosage of prokinetic agents firstly. Therefore, SBEN feeding was at the 7th ICU day. I think the optimal time for SBEN feeding may be on the 3rd to 7 days. Therefore, we will try to conduct a prospective to compare the survival benefit to start SBEN as early as possible (may be compared SBEN at 3rd & 7th ICU day.)

2. The discussion is somewhat short and should have more impact on the possibility of early bowel nutrition in patients with high NUTRICscore.

Response:

A. Thanks for your recommendation. We have modified the sentences in Lin 207-212 as Our observation that SBEN initiation on day 8 improves energy delivery could be a choice of feeding route for critically ill ICU patients with high nutrition risk. We also demonstrated SBEN could improve the feeding efficacy for malnutrition patients with high mNUTRIC score, therefore, further study is warrant to determine how early is possible to initiate SBEN if failure to achieve 60% of feeding goal in ICU. We suggest that for medical ICU patients with high mNUTRIC scores, adjuvant SBEN as early as possible is another option before shifting to parenteral nutrition.

B. We also addressed the limitation of our study in Lin 237-242 as “In addition, we don`t know how early is the optimal time to start SBEN by the retrospective observation cohort study. Therefore, a prospective study design is warrant. “

C. We try to emphysis the improtat of early SBEN in patients with high mNURTIC score as your suggestions. Therefore, we have modified the sentence in the discussion in Lin250-252 as “For medical ICU patients with high mNUTRIC scores, adjuvant SBEN as early as possible is another option before shifting to parenteral nutrition if gastric EN fails to achieve 60% of the energy goal

Thank you for your good suggestions and recommendations. We are pareparing to conduct a prospective interventional study to answer this important and interesting questions.

Reviewer 2 Report

Summary

This is an interesting as well as relevant study discussing the influence of postpyloric feeding in malnourished critically ill patients. The overall quality of this manuscript is very good. The manuscript is carefully and well-written and the graphics and tables add to the presentation.

Broad comments 

I have one major remark concerning a mismatch between study design and conclusion and a few minor remarks to improve the manuscript, which already has a good overall quality.

Specific comments 

·       Major remark:

o   Line 207: I cannot agree with that. The insertion of the tube happened late, only at one timepoint and only in cases with severe feeding intolerance. From this study design and cohort, one cannot pass judgement on the optimal timing of the postpyloric tube in critically ill patients.

·       Minor remarks

o   Line 99: That seems quite late (> 7 days) and was only applied in those patients with severe gastric feeding intolerance. Can you explain why this procedure was chosen?

o   Line 99: What measures were taken before and after SBEN to improve feeding tolerance?

o   Line 107: Why were APACHE and SOFA only measured until day 7 (when the transpyloric tube was inserted)?

o   Line 136: Albumin is not recommended as nutrition risk stratification tool.

o   Line 171: avoid page breaks within the table

o   Line 190: Please make the legend of your figure more obvious to the reader

Author Response

Major remark: 

Line 207: I cannot agree with that. The insertion of the tube happened late, only at one timepoint and only in cases with severe feeding intolerance. From this study design and cohort, one cannot pass judgement on the optimal timing of the postpyloric tube in critically ill patients.

Response: 

Agree with your opinion. We have modified the sentence as following:

“Our observation that SBEN initiation on day 8 improves energy delivery could be a choice of feeding route for critically ill ICU patients with high nutrition risk.”

Minor remarks

1. Line 99: That seems quite late (> 7 days) and was only applied in those patients with severe gastric feeding intolerance. Can you explain why this procedure was chosen?

Response: 

According to the ASPEN and the SCCM/ESPEN guidelines, the principle of feeding in critically ill patients as following:

Nutrition support therapy from early EN should be initiated within 24–48 hours in the critically ill patient who is unable to maintain volitional intake.

Early EN from gastric access should be used as the standard approach.

For patients who are unable to meet >60% of the energy and protein requirements by the EN alone, the use of supplemental parenteral nutrition (PN) should be considered after 7–10 days of EN.

Post-pyloric feeding should be used in patients with gastric feeding intolerance not solved with prokinetic agents.  

In our daily practice, we start to initial EN from gastric tube within 48 hours in ICU. The most of our patients suffered from septic shock and acute lung injury, therefore, the feeding amount were not good due to general edema and unstable of hemodynamic status in the first week. We will consult GI man at the 7th ICU day to evaluate and perform SBEN for those poor response to prokinetic agents and still in poor feeding status. Therefore, the time of SBEN insertion was 7-8th ICU day.

In fact, there are no any suggestions regarding when to do SBEN for those poor feeding patients.

2. Line 99: What measures were taken before and after SBEN to improve feeding tolerance?

In our daily practice, we try to improve the feeding status by three rules. 

Response: 

We will use prokinetic agents such as primperan 5mg iv q8h with/without  erythromycin iv 250 mg q12h to improve the feeding status in ICU. 

The feeding pump will also be applied for continuous feeding. 

The dietitian will adjust diet to semi-elemental formula. 

After SBEN insertion, most patients can stop primperan using. Some patients could be shifted to polymeric formula after SBEN.

3. Line 107: Why were APACHE and SOFA only measured until day 7 (when the transpyloric tube was inserted)?

Response: 

Our ICU routine will check APACHE II at the 1st ICU day and check SOFA at 1st, 3rd and 7th ICU day for every patient admitted to ICU. 

4. Line 136: Albumin is not recommended as nutrition risk stratification tool.

Response: 

Agree with your opinion. We showed the albumin level in the article due to every patient will be checked albumin level at the 1st ICU day. That is also a medical routine in our ICU.

5. Line 171: avoid page breaks within the table

Response: 

Thanks for your suggestion. Table 3 has been moved to Line 182 to avoid page breaks within the table. 

6. Line 190: Please make the legend of your figure more obvious to the reader

Response: 

We have modified the legend of figure 2 as: “Energy achievement rate > 65% was significantly associated with lower 30-, 60-, and 90-day mortality in patients with high nutrition risk.”

Reviewer 3 Report

Thank you for allowing me to review this manuscript.  I find it interesting and modern in application to clinical practice.  I have included comments below that will aid with clarity and presentation of the study findings.

In line 100, it was reported that patients diagnosed with severe malnutrition were with energy target achievement rates<30% by day of life 7.  Can you clarify if this diagnosis of “severe malnutrition” meets criteria based on current malnutrition recommendations or if this an arbitrary way you diagnosed it?

In Line 103, can you clarify that patients “at high nutrition risk” were determined to meet that criteria by mNUTRIC score?

I assume the amount of actual nutrition delivery is based on nursing documentation.  Please add this clarification to the manuscript.

From Table 2, can you explain why patients in the post-SBEN group received a large amount of volume (average of 2,845 ml/day), yet average calories were low at<1,000/day?

Please include more about your feeding protocol.  For example, how were feedings initiated and advanced before and after SBEN feedings?  The manuscript reports that the ICU uses a dietitian-approved feeding protocol but there was no detailed description of this.  Similarly, who decides when to advance feedings each day?  Does the dietitian recommend increased feeding rates to account for not meeting estimated nutrition needs (i.e. due to feedings being held for procedures, etc.)?

I appreciate the succinct, yet accurate major findings as indicated in the first paragraph of the Discussion section.

In your discussion, can you provide any insight as to why patients getting at least 65% of estimated nutrition needs have improved survival?  I find this interesting as even patients meeting 70-80% of their needs are still receiving inadequate nutrition.

Author Response

1. In line 100, it was reported that patients diagnosed with severe malnutrition were with energy target achievement rates<30% by day of life 7.

Can you clarify if this diagnosis of “severe malnutrition” meets criteria based on current malnutrition recommendations or if this an arbitrary way you diagnosed it?

Response: 

A. The definition of “severe malnutrition” in this research is according to the Academy of Nutrition and Dietetics and the American Society for Parenteral and Enteral Nutrition (A.S.P.E.N.). Nutr Clin Pract. 2013 Dec;28(6):639-50. 

B In the reference (Nutr Clin Pract. 2013 Dec;28(6):639-50.), severe malnutrition was defined as “Energy intake ≤50% for ≥5 days.”

C. In our populations, the energy intake rate was less than 30% in every patient at the 7th ICU day. Therefore, these populations fulfill the criteria of severe malnutrition. 

2. In Line 103, can you clarify that patients “at high nutrition risk” were determined to meet that criteria by mNUTRIC score?

Response: 

A. We have SOFA score at day 1, day 3 and day 7. Therefore, we have three mNURTIC scores for each patient. Patients enrolled into analysis should have at least 1 mNURTIC score ≧5.

B. We showed the mNURTIC score at the first ICU day in the article, not the highest mNURTIC score.

C. Therefore, all the patient enrolled into this study meet the criteria of “at high nutrition risk” by mNURTIC score. 

3. I assume the amount of actual nutrition delivery is based on nursing documentation. Please add this clarification to the manuscript.

Response: 

Yes. As you infer, the amount of actual nutrition delivery is based on nursing documentation.

We have added this sentence into materials and methods. “The amount of actual nutrition delivery is based on nursing documentation” (Line 112)

4. From Table 2, can you explain why patients in the post-SBEN group received a large amount of volume (average of 2,845 ml/day), yet average calories were low at<1,000/day?

Response: 

A. To answer this question, we have checked the raw data.

B. We misprint the data, 2,845ml were the "Sum" feeding volume of 1 to 3 days after SBEN, not the average volume per day. Therefore, the accurate average feeding volume of 1 to 3 days after SBEN were 948ml/day, average calories were 907.2/day

C. We will provide our raw data (excel file) to the reviewer.We have corrected the wrong data in table 2 and in results. 

5. Please include more about your feeding protocol.  For example, how were feedings initiated and advanced before and after SBEN feedings?  

Response: 

A. The volume-based feeding protocol has been implemented in TVGH’s ICU since June 2015 as our previous report. (Nutrients. 2017 May 21;9(5). pii: E527.) 

B. This protocol has been slightly modified from the PEP uP protocol (Crit. Care Med. 2013, 41, 2743–2753).

C. After SBEN insertion, most patients could be shifted to polymeric formula 

6. The manuscript reports that the ICU uses a dietitian-approved feeding protocol but there was no detailed description of this.  Similarly, who decides when to advance feedings each day? 

Response: 

In our ICU, the advanced feedings will be determined by team discussion in the morning during ward round. The team members including attending physician (ICU staff), ICU fellow, resident, duty nursing and dietitian. The final decision will made by the team leader “attending physician” after team discussion.

7. Does the dietitian recommend increased feeding rates to account for not meeting estimated nutrition needs (i.e. due to feedings being held for procedures, etc.)?

Response: 

Yes. The dietitian would not only let team members know the nutrition goal of patients, but also would recommend the adequate feeding rates. 

If feeding is interrupted for any reason, the dietitian will calculate the NPO duration and adjust feeding rate by checking the feeding rate table. (Maximal rate: 150 cc/h). 

The dietitian also adjust diet to easier digestion formula or condense formula to help patients to achieve their feeding goal.

8. I appreciate the succinct, yet accurate major findings as indicated in the first paragraph of the Discussion section.

Response:  Thank you.

9. In your discussion, can you provide any insight as to why patients getting at least 65% of estimated nutrition needs have improved survival?  I find this interesting as even patients meeting 70-80% of their needs are still receiving inadequate nutrition.

Response:

A. Our previous studies show that high–nutrition-risk ICU patients who receive at least 65% of their energy requirements and consume at least 800 kcal/day have significantly lower in-hospital, 14-day, and 28-day mortality rates (Reference #1 and Reference#9).

B. In this cohort, the 65% was calculated by ROC curve to have the best AUC. Due to a retrospective design and not a mechanism- based study, we could only mention this finding but very difficult to explant the reason.

C. In fact, a previous study has shown that a U-shaped curve was significant association with mortality. (Crit Care. 2016 Nov 10;20(1):367.) The group suggested the lowest mortality was noted at 70 % of estimated nutrition needs. 

D. AdCal/REE from 0 % to 70 % was associated with decreasing mortality (HR= 0.98, 95 % CI 0.97–0.99), while an AdCal/REE ratio ≥70 % was associated  with increasing mortality (HR= 1.01, 95 % CI 1.01–1.02).
